# Symmetric Dense Inception Network for Simultaneous Cell Detection and Classification in Multiplex Immunohistochemistry Images

**Anonymous**                                                    ANONYMOUS EMAIL

**Editor:** Editor's name

## Abstract

Deep-learning based automatic analysis of the multiplex immunohistochemistry (mIHC) enables distinct cell populations to be localized on a large scale, providing insights into disease biology and therapeutic targets. However, standard deep-learning pipelines performed cell detection and classification as two-stage tasks, which is computationally inefficient and faces challenges to incorporate neighbouring tissue context for determining the cell identity. To overcome these limitations and to obtain a more accurate mapping of cell phenotypes, we presented a symmetric dense inception neural network for detecting and classifying cells in mIHC slides simultaneously. The model was applied with a novel stop-gradient strategy and a loss function accounted for class imbalance. When evaluated on an ovarian cancer dataset containing 6 cell types, the model achieved an F1 score of 0.835 in cell detection, and a weighted F1-score of 0.867 in cell classification, which outperformed separate models trained on individual tasks by 1.9% and 3.8% respectively. Taken together, the proposed method boosts the learning efficiency and prediction accuracy of cell detection and classification by jointly learning from both tasks.

**Keywords:** Deep learning, Digital pathology, Multiplex immunohistochemistry

## 1. Introduction

Multiplex immunohistochemistry (mIHC) is an important technique to resolve the spatial arrangement of multiple cell phenotypes within the tissue, which has been used to elucidate biological processes and predict therapeutic outcomes (Zahir et al., 2020), (Lu et al., 2019). A promising utility of mIHC is to locate the programmed cell death-1 (PD-1) and assess its correlation with CD8+, CD4+ and FOXP3+ tumour-infiltrating lymphocytes, which are of great interest to pathologists given the role of PD1 as an immune checkpoint protein and an important target of immunotherapy (Diana et al., 2016), (Halse et al., 2018).

To automatically identify the expression of antigens, several deep learning models have been developed to detect and classify cell types in mIHC images (Hagos et al., 2019, 2021; Narayanan et al., 2021). However, these methods were designed to process the two tasks separately, which has several limitations. Firstly, training and predicting with separate models is costly in computing time and resources, which constrains the usage of models in large scale datasets (Song et al., 2019). Moreover, classifying cells with limited local context features might lead to a reduction in performance. For example, assembling predictions from multiple neighbouring regions has been shown to produce more accurate results than

classification performed on the central region alone (Sirinukunwattana et al., 2016). On the other hand, the cell detection model without considering class-wise abundance was shown to miss out on rare cell types in an unbalanced dataset (Hagos et al., 2021). As a result, models combining features from cell detection and classification are desired.

Recently, several efforts have been made to simultaneously detect and classify cells on H&E images. Song et al. (2019) proposed a synchronized asymmetric deep-learning framework to parallelly detect and classify cells in bone marrow specimens. Graham et al. (2019) constructed a three-branch network to simultaneously segment and classify nuclei on histology images. However, both strategies required exhaustive manual annotations of nuclei boundary. Such tedious and time-consuming annotations might not be essential for cell classification in mIHC samples, where distinct types of cells were readily marked by chromogenic staining. In fact, Fassler et al. (2020) has demonstrated that a cell segmentation model trained on arbitrary cell masks generated using only annotations of cell centres could achieve desired performance in mIHC images.

Despite that cells are distinguishable by the expression of markers, it is still nontrivial to identify cell types on mIHC slides. Specifically, some cell types may express multiple markers, and some exhibit high variability in staining intensities. One solution to overcome the intermix of staining is to deconvolve colours into separate channels, then identify cells based on the combined positive or negative signal in different channels (Fassler et al., 2020; Blom et al., 2017; Chen and Srinivas, 2015; Duggal et al., 2017; Abousamra et al., 2020; Lahiani et al., 2018). However, such a strategy requires prior knowledge about the color range associated with each marker, which limits its application in cases where staining colour spans a broad spectrum. Moreover, methods relying on colour deconvolution and segmentation of homogeneous colours are insufficient to identify individual cells in close proximity, which compromise the accuracy of spatial analysis involving distributions of single cells (Fassler et al., 2020; Lahiani et al., 2018).

To overcome these shortcomings of previous methods, we proposed a neural network to detect and classify cells simultaneously on mIHC slides. We compared the performance of different network structures and evaluated the dependency between detection and classification tasks. The proposed method outperforms baseline models trained separately on cell classification and detection. To the best of our knowledge, this is the first end-to-end solution for detecting and classifying cells in mIHC images.

## 2. Material and Methods

### 2.1 Dataset and annotations

Our dataset contained 9 whole slide images of high-grade serious ovarian cancer stained for CD8, CD4, FOXP3 and PD1. To train and validate the proposed method, a total of 3674 single cell annotations were collected from 9 slides with experts putting colourful dots at the cell centre to label 6 dominant cell types distinguished by expressions of markers. For sake of brevity, we denoted the 6 cell types as CD8+, CD4+FOXP3-, CD4+FOXP3+, PD1+CD4-CD8-, PD1+CD4-CD8+, and PD1+CD4+CD8-. Antigens expressed by each cell type were detailed in Table 1. Examples of the annotated cells were shown in Figure 1b. A total of 257 image patches containing annotations were extracted from 4 slides and randomly split

Table 1: Number of cells in training, validation, and test datasets.

| Antigen expression | | | | Cell types | Training | Validation | Test |
|---|---|---|---|---|---|---|---|
| PD1 | CD8 | CD4 | FOXP3 | | | | |
| - | + | - | - | CD8+ | 110 | 33 | 344 |
| - | - | + | - | CD4+FOXP3- | 380 | 137 | 515 |
| - | - | + | + | CD4+FOXP3+ | 192 | 110 | 368 |
| + | - | - | - | PD1+CD4-CD8- | 33 | 26 | 21 |
| + | + | - | - | PD1+CD4-CD8+ | 453 | 183 | 612 |
| + | - | + | - | PD1+CD4+CD8- | 50 | 33 | 74 |

with a 7:3 ratio into training (180) and validation (77) datasets. Cells from the other 5 slides were used for testing. The composition of each dataset was described in Table 1.

## 2.2 Training data preparation

Slides were scanned at 40x magnification and rescaled to 20x with a resolution of 0.44 µm/pixel. Regions with cell annotations were cropped into $224 \times 224$ patches $I \in \mathbb{R}^{224 \times 224 \times 1}$ with a stride of 120 pixels. For classification, we generated a binary mask per cell type $M_k \in \mathbb{R}^{224 \times 224 \times 1}$ by marking a circle area with a radius $r = 2$ pixels centred at each dot annotation. The distance threshold $r$ was set empirically to cover a considerable cell area while avoiding the overlap of adjacent annotations. $M_k$ for a total of 6 cell classes were stacked into a classification mask $M_c \in \mathbb{R}^{224 \times 224 \times 6}$. We generated the detection mask $M_d \in \mathbb{R}^{224 \times 224 \times 1}$ by taking the maximum values of $M_c$ channel-wise, and a background mask $M_b \in \mathbb{R}^{224 \times 224 \times 1}$ as the inversion of $M_d$. The detection mask labelled locations of all cells regardless of cell types, and the background mask represented tissue regions without identified cells. We added the background mask to $M_c$ as the $7^{th}$ channel. The $i^{th}$ input in the training dataset was represented as $(I^i, M_c^i, M_d^i)$. Data augmentation was performed by randomly flipping the input horizontally and vertically.

## 2.3 Network architecture and training

We proposed a fully convolutional network Symmetric Distance Regularized Dense Inception neural Network (S-DRDIN) to simultaneously detect and classify 6 immune cell types in the mIHC images. The network structure was built on the original DRDIN (Narayanan et al., 2021), with modifications to the decoder to enable parallel predictions for cell classes and locations.

As shown in Figure 1, the network was constructed following a U-net structure (Ronneberger et al., 2015) with inception blocks (Szegedy et al., 2015) as the basic convolution module. The encoder comprised 4 inception blocks linked by $2 \times 2$ averaged pooling layers to downsample features by a factor of 2 after each convolutional step. In comparison with the original DRDIN which comprised a single-branch decoder, S-DRDIN has two decoders to predict for class map and detection map simultaneously.

As suggested by Song et al. (2019), cell classification relied on less information than detection. To reduce the potential deleterious impact on classification induced by redundant information learnt from the detection task, we applied stop-gradient to all the skip

connections between the encoder path and the detection branch (Figure 1). This operation prohibited the direct gradient transfer from the detection branch to the encoder, thereby disentangled the detection and classification information for low-level features and encouraged the network to optimize the feature map for the cell classification task. To compensate for the potential loss of useful information caused by the stop-gradient operation, outputs of the last inception blocks of the classification branch was incorporated into the detection branch, followed by two convolution layers with $3 \times 3$ kernels and a final convolution layer with 1x1 kernels to generate the detection output. For the classification branch, the last convolution layer was immediately added after the last inception block to produce an output of size $224 \times 224 \times 7$. Relu activation was applied to all the convolution layers except for the last convolution layers of classification and detection branches, which were activated with Softmax and Sigmoid respectively. The model was trained for 100 epochs with a batch size of 4. Weights were initialized using uniform glorot (Glorot and Bengio, 2010) and optimized using Adam (Kingma and Ba, 2015) with a learning rate of $10^{-3}$.

### 2.4 Loss function for cell classification and detection

We used cross-entropy to calculate loss between the classification branch output and class map. To tackle the imbalance of class occurrences, the loss of each pixel was assigned with a weight $w_c$ calculated as the inverse proportion of the number of pixels of the corresponding class in the batch, as given by,

$$w_c = \frac{N \cdot N_c \cdot B}{\sum_{i=1}^{N} x_i^c} \tag{1}$$

where $N_c$ and N denoted the number of channels and number of pixels in each channel of the class map input. B is the batch size and $x_i^c$ is the pixel value in channel $c$. The pixel-weighted cross-entropy loss function is defined as

$$L_{class} = -\frac{1}{N_c \cdot N} \sum_{c=1}^{N_c} \sum_{i=1}^{N} w_c \cdot \log(p_c(x_i)) \tag{2}$$

where $p_c(x_i)$ denotes the predicted probability from the classification branch after the Softmax activation. We separately calculated the binary cross-entropy for the detection branch. The overall model was trained in an end-to-end fashion with constraints from both cell detection and classification. The total loss is defined as,

$$L_{total} = L_{class} + L_{detect} \tag{3}$$

The model trained with $L_{total}$ is referred to as the proposed S-DRDIN. To evaluate the intra-influence between classification and detection tasks, we also tested models trained on detection and classification separately and trained without the stop-gradient operation. The three model variants are defined as follows,

1. DRDIN-detection: does not contain the classification branch

2. DRDIN-classification: does not contain the detection branch

3. S-DRDIN-gradient: allows back-propagation for all connections between encoder layers and the detection branch

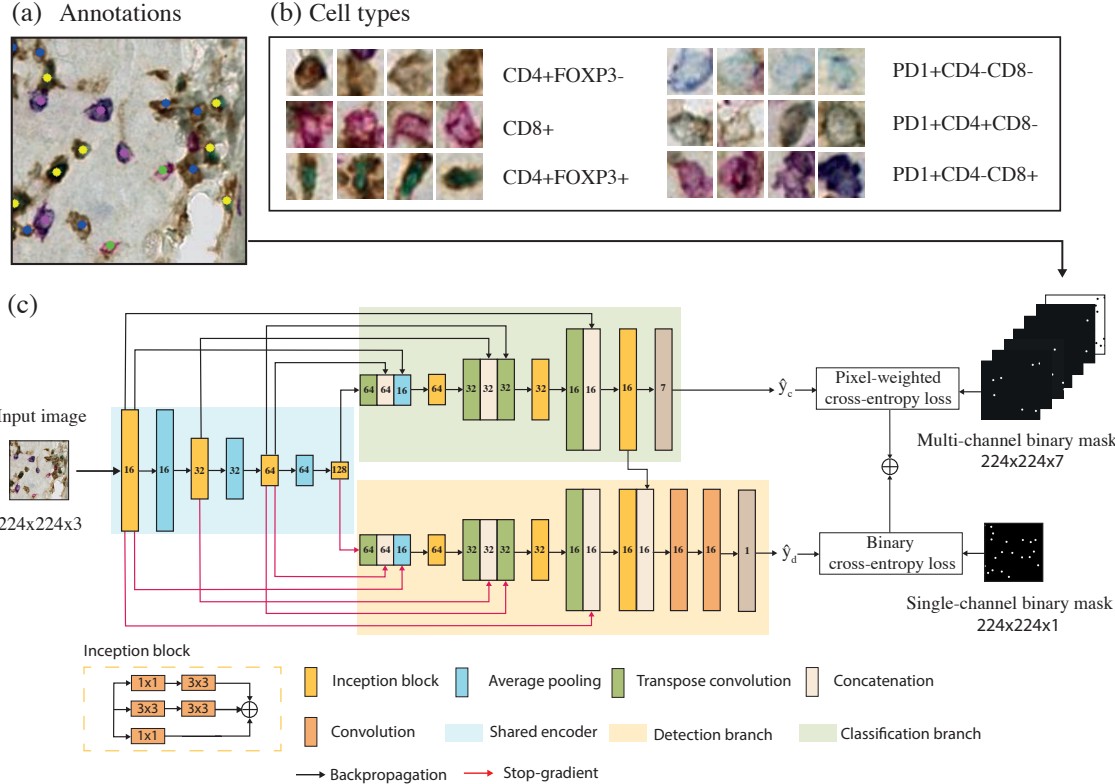

Figure 1: Framework of the proposed pipeline.

## 2.5 Post-processing and evaluation

To obtain the locations of detected cells on the slide, we applied a post-processing pipeline to the probability map generated from the detection branch. Firstly, a threshold optimized for the F1-score of each model was applied to binarize the predicted detection mask. Next, connected components with an area smaller than 50 were discarded to remove noise in the background. Then we calculated the distance transform and identify the local maxima with a sliding window of size $15 \times 15$. The local maxima were dilated by a disk with a radius of 2 pixels. Lastly, centres of the instances were recorded as detected cell locations.

For cell classification, predicted values from each of the 6 cell-type channels were averaged for the $49 \times 49$ square region surrounded each identified cell. The predicted cell class $k$ was determined by

$$\hat{k} = \operatorname*{argmax}_{k} \frac{1}{N} \sum_{i=1}^{N} p_c^k(x_i) \tag{4}$$

where $p_c^k(x_i)$ represents the network output for pixel $x_i$ at the channel corresponding to class k. N is the total number of pixels within the square region surrounding the detected cell. For robust evaluation of model performance, we trained each model separately on 3 datasets with training and validation split at different randomized states. The average model

Table 2: Cell detection performance evaluated for different models.

| Methods | Precision | Recall | F1-score | True positives | False positives |
|---|---|---|---|---|---|
| U-Net | $0.848 \pm 0.011$ | $0.772 \pm 0.016$ | $0.808 \pm 0.003$ | $1493 \pm 30$ | $268 \pm 28$ |
| CONCORDe-Net | $0.921 \pm 0.037$ | $0.660 \pm 0.017$ | $0.768 \pm 0.001$ | $1276 \pm 33$ | $111 \pm 58$ |
| DRDIN-detection | $0.851 \pm 0.017$ | $0.783 \pm 0.021$ | $0.816 \pm 0.004$ | $1515 \pm 41$ | $266 \pm 43$ |
| S-DRDIN-gradient | $0.783 \pm 0.098$ | $\mathbf{0.809 \pm 0.039}$ | $0.793 \pm 0.032$ | $1565 \pm 76$ | $456 \pm 274$ |
| S-DRDIN | $\mathbf{0.941 \pm 0.016}$ | $0.750 \pm 0.031$ | $\mathbf{0.835 \pm 0.013}$ | $1450 \pm 61$ | $92 \pm 30$ |

performance was reported for the 5 hold-out testing slides containing expert annotations on every identifiable cell within a given region. A predicted cell inside the region was considered as true positive if it fell within 10 pixels to an expert annotation, otherwise false positive. The false negative was counted as the number of annotated cells missed out by the model. The precision, recall and F1-score were reported for cell detection evaluation. Performance of cell classification was assessed for all annotated cells detected by a model, with precision, recall and F1-score weighted by the proportion of cell types computed for model comparisons.

## 3. Results

### 3.1 Cell detection performance

We evaluate the detection performance for different variants of the proposed S-DRDIN, and compared it against other states of the art U-Net (Ronneberger et al., 2015) and CONCORDe-Net (Hagos et al., 2019). The proposed S-DRDIN obtained an F1 score of 0.835, which was 2.7% and 6.7% higher than the U-Net and the CONCORDe-Net respectively. It also outperformed DRDIN-detection and S-DRDIN-gradient by 1.9% and 4.2% (Table 2). The improvement was mainly attributed to the increase in precision and the reduction in false positives, while S-DRDIN was less sensitive to true positive cells as compared to U-Net, DRDIN-detection and S-DRDIN-gradient, as reflected by the lower recall (Table 2). Specifically, the model tended to overlook CD4+FOXP3- and PD1+CD4+CD8- cells, with both classes showing a detection recall lower than 0.6 (Table A.2). These results were not fully explained by the data imbalance, as CD4+FOXP3- was the second most abundant class, and the averaged recall obtained for PD1+CD4-CD8- cells was 0.833 despite it being the rarest cell type Table 1. It is likely that the staining colour for CD4+FOXP3- and PD1+CD4+CD8- makes them less distinguishable from the tissue background as compared to other cell types (Figure A.1).

Interestingly, the S-DRDIN-gradient model, which jointly learned from classification and detection without stopping backpropagation for the detection branch, achieved the best recall but performed worse than the DRDIN-detection concerning the F1-score. This observation was consistent with previous findings in Song et al. (2019) where the intra-influence between detection and classification reduces performance for both tasks. Here we demonstrated the stop-gradient operation as an effective approach to reduce conflicts between the two tasks and restore the detection accuracy. A potential explanation was that blocking the gradient flow from detection branch derived feature maps more representative

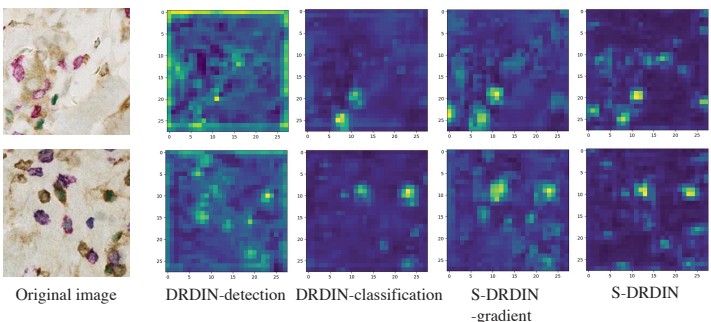

|  |  |  |  | |
|---|---|---|---|---|
| Original image | DRDIN-detection | DRDIN-classification | S-DRDIN -gradient | S-DRDIN |

Figure 2: A qualitative comparison of feature maps generated by the last layer of the encoder of different models. Yellow indicates high value.

for cell class identities, which limited the detection for cells with ambiguous class identity, therefore generated a more precise cell map.

To better understand the contribution of stop-gradient operation for cell detection, we qualitatively compared the feature maps produced by the last layer of the encoder of different models. As shown in Figure 2, activation produced by S-DRDIN was more localized to cell regions as compared to S-DRDIN-gradient and DRDIN-detection, which was consistent with our hypothesis that stop-gradient functioned in suppressing the redundant information introduced by the backpropagation of the detection branch. The activation map indicated that these features might relate to the false positives in tissue background. Additionally, both S-DRDIN and S-DRDIN-gradient showed activating signals for a larger number of cells than the DRDIN-classification, suggesting the advantages of additional information introduced by parallelly learning from detection and classification.

## 3.2 Cell classification evaluation

We compared performance for cell classification among different network and training strategies. Both the U-Net and DRDIN-classification were trained on the 7-channel binary masks described in Section 2.2, with U-Net trained without additional weights assigned to the loss function. Classification for the CONCORDe-Net was performed using an SCCNN classifier (Sirinukunwattana et al., 2016) trained on single-cell patches extracted from the annotations. S-DRDIN achieved the highest score for weighted precision (0.870), weighted recall (0.872) and weighted F1-score (0.867) among all the five models, including U-Net, Concordent-Net and DRDIN-classification that were trained separately on detection and classification (Table 3). The training time of S-DRDIN was shorter than models trained on discrete tasks except for the U-Net, which was due to the difference in parameters (Table A.1). Predictive accuracy of CD8+, PD1+CD4-CD8- and PD1+CD4+CD8- was sub-optimal compared to that of the other cell type, which was likely due to the limitation of sample size (Table A.2). On the other hand, S-DRDIN-gradient also outperforms the models trained on separate tasks, suggesting that the incorporation of cell location information into the training stage could improve cell classification performance. Additionally, the employment of the pixel-weighted cross-entropy increased the weighted F1-score by at

Table 3: Cell classification performance of different methods.

| Methods | Weighted Precision | Weighted Recall | Weighted F1-score |
|---|---|---|---|
| U-Net | $0.768 \pm 0.006$ | $0.795 \pm 0.002$ | $0.761 \pm 0.006$ |
| CONCORDe-Net | $0.797 \pm 0.004$ | $0.816 \pm 0.009$ | $0.778 \pm 0.011$ |
| DRDIN-classification | $0.839 \pm 0.011$ | $0.834 \pm 0.009$ | $0.829 \pm 0.018$ |
| S-DRDIN-gradient | $0.860 \pm 0.015$ | $0.860 \pm 0.018$ | $0.857 \pm 0.021$ |
| S-DRDIN | $\mathbf{0.870 \pm 0.011}$ | $\mathbf{0.872 \pm 0.009}$ | $\mathbf{0.867 \pm 0.015}$ |

least 5% than the baseline models, demonstrating the efficiency of pixel wighting strategy for reducing the negative influence of data imbalance. Examples of cell detection and classification results were shown in Figure A.1. Noted that S-DRDIN was the only model which successfully identified the rare PD1+CD4-CD8- cell in the region.

## 4. Discussion

Our study aims at designing a deep-learning method for detecting and classifying cells simultaneously in mIHC samples. The proposed model exploited the mutual dependency of the two highly relevant tasks, and introduced a novel stop-gradient approach to reduce the conflict between detection and classification features. In comparison with the conventional deep learning pipeline that processed the two tasks as separate steps, the proposed model directly predicts locations of cells of different types on a given mIHC image, therefore reduces the computation costs and speed up the analysis for large scale datasets. Moreover, we demonstrated that the combination of stop-gradient operation, pixel-weighted cross-entropy loss and parallel learning from detection and classification results in higher precision for detection as well as better performance for classification.

This study was limited by the small dataset of 3674 single cell annotations with an extreme class imbalance. In future work, we intend to evaluate the generalization of the model on larger multiplex image datasets from different sources. Also, additional modifications can be applied to the model architecture to improve the detection recall. Once fully developed, the method can accelerate the image analysis for large cohorts and promote the understanding of the spatial relationship between diverse components of the tumour immune microenvironment.

## 5. Conclusion

We presented a network S-DRDIN for simultaneous cell detection and classification in mIHC images. With the aid of stop-gradient and a loss function accounted for class proportions, the proposed model achieved an F1 score of 0.835 in cell detection, and a weighted F1 score of 0.867 in classification, which were 1.9% and 3.8% higher than the model trained separately on individual tasks.

Table A.1: Training time of different methods for 100 epochs.

| Methods | Training time |
|---|---|
| U-Net-detection+classification | 0.09 hours |
| Concordenet-detection+classification | 0.73 hours |
| DRDIN-detection+classification | 0.16 hours |
| S-DRDIN | 0.12 hours |

Table A.2: Cell classification performance and detection recall of the proposed S-DRDIN for different cell types. Bold values highlight scores lower than 0.8.

| Cell types | Precision | Recall | F1-score | Detection Recall |
|---|---|---|---|---|
| CD8+ | $0.833 \pm 0.025$ | $\mathbf{0.683 \pm 0.069}$ | $\mathbf{0.749 \pm 0.032}$ | $\mathbf{0.731 \pm 0.113}$ |
| CD4+FOXP3- | $0.910 \pm 0.059$ | $0.954 \pm 0.014$ | $0.931 \pm 0.024$ | $\mathbf{0.525 \pm 0.103}$ |
| CD4+FOXP3+ | $0.967 \pm 0.002$ | $0.974 \pm 0.019$ | $0.970 \pm 0.009$ | $0.865 \pm 0.017$ |
| PD1+CD4-CD8- | $\mathbf{0.675 \pm 0.035}$ | $\mathbf{0.750 \pm 0.039}$ | $\mathbf{0.711 \pm 0.037}$ | $0.833 \pm 0.034$ |
| PD1+CD4-CD8+ | $0.848 \pm 0.032$ | $0.911 \pm 0.018$ | $0.877 \pm 0.009$ | $0.814 \pm 0.062$ |
| PD1+CD4+CD8- | $\mathbf{0.646 \pm 0.065}$ | $\mathbf{0.468 \pm 0.343}$ | $\mathbf{0.514 \pm 0.264}$ | $\mathbf{0.527 \pm 0.019}$ |

**Appendix A.**

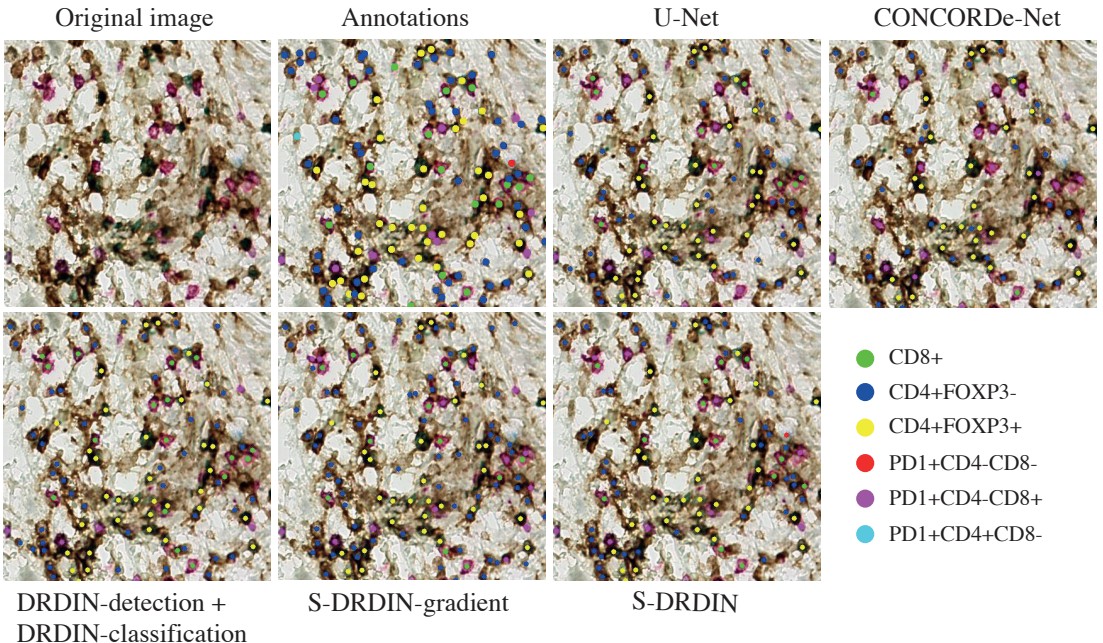

Figure A.1: Examples of cell detection and classification outputs from different methods.

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
