# OpenReview forum: "Symmetric Dense Inception Network for Simultaneous Cell Detection and Classification in Multiplex Immunohistochemistry Images"
_MICCAI.org/2021/Workshop/COMPAY — COMPAY 2021_

### Official Review · Reviewer_Jw3G · 2021-08-06
**A novel fully-supervised method for joint detection and classification, validated on an extremely small mIHC data set**

**Rating:** 6
**Confidence:** 3

**Review:**

The paper is well written, and benchmarks the novel solution against multiple state-of-the-art approaches. The achieved cell detection and classification accuracies outperform existing methods in the provided data sets.
The paper suffers from the limited data set (9 slides) which could introduce a bias and does not allow conclusions about the generalization capabilities of the method. Second, neither inter-observer accuracies are provided, nor confidence intervals for the used metrics (f1, prec, recall, tp, fp). Last, mIHC methods are sensitive to the specific assay, sample preparation and imaging, therefore stain normalization and/or data augmentation in color space could play an important role when translating the method from theory to clinical routine, unfortunately these considerations were outside of the scope of this study.

---

### Official Review · Reviewer_H2RY · 2021-08-22
**In this paper, the authors proposed a pipeline for cell detection and classification in mIHC images. They proposed useful methods to boost the performance of the pipeline including using stop-gradient operation, applying pixel-weighted cross-entropy loss and parallel learning from detection and classification.**

**Rating:** 8
**Confidence:** 5

**Review:**

The paper is written clearly. There some some corrections required to be done to improve the read of the manuscript as listed bellow.

- Rephrasing required for these lines since it is not very clear: “However, both strategies required exhaustive … distinct morphology”.
- The authors could specify the unit of counts presented in Table1. Was it cells? Image patches? Or something else?
- Authors could explain the importance of each term they have used to introduce their model (Symmetric, Distance Regularized, Dense, Inception, neural Network) for the aimed tasks.
- Since the authors have done 5FCV, the results reported in the tables should be in the form of mean+/-std rather than a value only.
- The authors have claimed that their proposed approach reduces the computation costs and speeds up the analysis. However, no information with regards to computation costs and speed of various methods is presented in the manuscript. This information needs to be included.
- The authors could present some bad results along with the discussion of the outcome.

---

### Decision · Program_Chairs · 2021-08-25

Accept